# Newborn Hearing Screening and Intervention in Children with Unilateral Hearing Impairment: Clinical Practices in Three Nordic Countries

**DOI:** 10.3390/jcm10215152

**Published:** 2021-11-02

**Authors:** Nina Jakhelln Laugen, Elsa Erixon, Kerttu Huttunen, Elina Mäki-Torkko, Ulrika Löfkvist

**Affiliations:** 1Department of Psychology, Norwegian University of Science and Technology (NTNU), 7491 Trondheim, Norway; 2Department of Surgical Sciences, Uppsala University, 75185 Uppsala, Sweden; elsa.erixon@akademiska.se; 3Research Unit of Logopedics and Child Language Research Center, Faculty of Humanities, University of Oulu, 90014 Oulu, Finland; kerttu.huttunen@oulu.fi; 4Department of Otorhinolaryngology, Head and Neck Surgery, Oulu University Hospital, 90220 Oulu, Finland; 5Medical Research Center Oulu, University of Oulu, 90014 Oulu, Finland; 6Audiological Research Centre, Faculty of Medicine and Health, 70182 Örebro, Sweden; Elina.maki-torkko@oru.se; 7School of Medical Sciences, Faculty of Medicine and Health, 70182 Örebro, Sweden; 8Department of Public Health and Caring Sciences, Uppsala University, 75122 Uppsala, Sweden; ulrika.lofkvist@pubcare.uu.se; 9Department of Clinical Science, Intervention and Technology (CLINTEC), Karolinska Institute, 17177 Stockholm, Sweden

**Keywords:** unilateral hearing loss, single-sided deafness, hearing screening, follow-up, habilitation, healthcare inequality, early intervention, family-centred intervention

## Abstract

Studies have limitedly considered children with early-identified unilateral hearing impairment (UHI), and clinical practices regarding screening, diagnostics and habilitation in this group are rarely documented. In this study, routines for newborns with UHI from screening to diagnostics and habilitation were explored in Norway, Sweden and Finland. An online survey was sent to hospitals responsible for the hearing diagnostics of children requesting information about their practices regarding congenital UHI. Responses covered 95% of the children born in the three included countries. The results revealed large variations in ways of organising healthcare and in clinical decisions regarding hearing screening, diagnostics and habilitation of children with congenital UHI. Finally, implications for policy making and research are also discussed.

## 1. Introduction

Among the types of hearing impairment (HI), unilateral hearing impairment (UHI) is relatively common, comprising around 20–50% of all congenital HIs [1,2,3,4]. Still, compared to especially severe and profound bilateral HI, research on the outcomes of congenital or early-onset UHI is scarce [5], and recommendations for the timing, scope and content of early intervention for this group are based on limited empirical evidence [6]. This allows much room for individual practices, and clinics may vary in their approaches to diagnostic procedures and habilitation.

Universal newborn hearing screening has been implemented in Norway, Sweden and Finland for up to 20 years, and early intervention has been available for children with HI and their families. The actual practices in screening, diagnostic routines and habilitation are not registered or monitored by, for instance, healthcare authorities in these countries and, therefore, remain ill-documented. For children with UHI, the lack of evidence-based recommendations or guidelines might increase the risk of unequal access to services, and the lack of documentation of clinical practices limits the possibilities of addressing these differences in policy making and healthcare planning. Therefore, this study explores the status of screening, diagnostics and intervention practices in Norway, Sweden and Finland.

### 1.1. Definitions, Prevalence and Outcomes

The World Health Organization (WHO) defines UHI in adults as a pure tone average (PTA) of hearing thresholds at 0.5, 1, 2 and 4 kHz, being 35 dB or greater in the worse ear and <20 dB in the better ear [7]. For children, the WHO does not provide any definition, and those used in empirical studies are diverse, with PTAs being calculated over various frequencies and with variations in PTA thresholds, both in the worse and better ears. Reported prevalence of early-identified UHI ranges from 0.6 to 1 per 1000 newborns [3,4,8,9]. Additionally, more than half of all UHI cases may present themselves later in childhood [4], and over time, 20% to 40% of all congenital UHI cases progress into bilateral HI [4,9].

UHI reduces the capacity for binaural hearing and thereby the ability to listen in noise and localise sounds [10,11,12]. Vestibular functions (balance) may also be impaired in children with UHI [13]. Reduced listening performance and associated psychosocial difficulties, such as reduced social participation and lower quality of life, have been reported in adults with UHI [12,14,15,16].

It is reasonable to assume that children experience similar difficulties as adults. For example, some studies have suggested associations between UHI and lower quality of life in school-age children [17,18,19]. Moreover, UHI might also be associated with delays in some areas, such as the development of early auditory skills, spoken language [20] and cognition [21,22,23]. However, other studies have reported cognitive, auditory and language development of children with UHI to be similar to normal hearing age peers [24,25].

Heterogeneity in reported outcomes can be attributed to various factors, such as methodological issues and variations in the study samples regarding aetiology and access to early intervention [24,26,27,28]. As shown by recent systematic reviews [5,28], only a few high-quality studies have addressed development in children with early-identified UHI. Overall, the dearth of studies and inconsistency of outcomes leave clinicians with limited evidence base, which, in turn, may induce heterogeneity in routines regarding diagnosis and habilitation [27].

### 1.2. Healthcare Systems in Norway, Sweden and Finland

Norway, Sweden and Finland share many common features in culture, social structures and geography. They are all relatively small countries, Sweden having just over 10 million inhabitants and Finland and Norway just over 5 million each. Their healthcare systems are largely tax-funded and accessible for all inhabitants at minimal or no cost, but a relatively large share of the population lives in sparsely populated areas where access to medical or educational expertise might require some travelling. In the three countries, all hospitals with a birth unit perform hearing screening, whereas a more limited number of hospitals have clinics responsible for the diagnostics and habilitation of HI. Children who do not pass the screening are referred to one of these hospitals. Diagnosing hospitals vary in size [29,30,31].

In all the Nordic countries, newborn hearing screening was introduced stepwise, starting in late 1990s in a limited number of hospitals followed by an increasing number over the next ten years. Coverage is not reported in national statistics, although estimations and data between 95% and 100% have been reported by individual hospitals and in surveys [32,33,34,35]. In Norway, national guidelines for screening, diagnosis and habilitation of children with HI exist. There are specific recommendations for children with UHI, such as information to parents, audiological monitoring every six months and considering prescription of hearing aids for mild and moderate HI, up to 70 dB in the affected ear [36]. In Finland, guidelines exist only for children with bilateral hearing aids [37], whereas Sweden lacks national guidelines. Overall, hearing-related healthcare is fairly accessible in the three countries, although geographical distances may impede some regions, and available resources may depend on the hospital’s size.

### 1.3. Best Practice and Clinical Decision Making

Alongside the implementation of hearing screening and a growing body of research on early-identified HI in general, guidelines have been developed for hearing screening, diagnostics and early intervention. The position statements of the Joint Committee of Infant Hearing [38,39,40,41] and the consensus statement on family-centred early intervention [42] agree on the central aspects of the best practices for early hearing detection and intervention, including universal screening, early diagnosis and access to early intervention. Family-centred early intervention is emphasised, such as parent–professional partnerships, family empowerment and informed choice [39,42].

Best-practice recommendations typically target all types of HI, including UHI. Bilateral screening is recommended where possible [43]. There is widespread consensus that children with any type or degree of HI should receive targeted and appropriate services [39,42]. Still, the content and scope of the appropriate intervention is unspecified. In a recent review addressing the importance of high-quality hearing screening programmes, screening is suggested to cover at least one ear [44], thus leaving some uncertainty as to whether intervention for children with UHI should be prioritised or not.

Recently, early intervention for UHI has received increased attention [21]. In Canada, specific recommendations for UHI management have been published [6,45]. Some studies have suggested improved auditory functioning if amplification is provided at an early age for children with UHI [46]. A systematic review on auditory outcomes after providing amplification [47] reported heterogeneous results, and, importantly, none of the studies included in the review concerned children younger than school-age. Furthermore, there is a lack of research that investigates the effect of early intervention beyond audiological management, such as special education, family counselling and speech and language therapy, in children with UHI.

Given the current state of knowledge, it is not surprising that various practices are presented in existing literature. For example, although most screening programmes perform screening in both ears, many hospitals in Finland and Switzerland currently screen only one ear and do not take further action if that ear passes [48]. This poses a risk of missing many UHI cases. Furthermore, access to early intervention programmes may vary. In the United States, eligibility criteria for early intervention differ between states, resulting in the inclusion of children with UHI in some states only [49]. Amplification is often recommended late [27], and professionals may vary in whether they recommend amplification at all, especially in mild UHI cases [50]. Moreover, parents may be reluctant to amplification or early intervention, as the benefit might not be as evident for children with UHI compared to more profound degrees of bilateral HI [51,52]. Overall, the risk of unequal access to hearing healthcare is evident, and some children might not receive the appropriate support. Although practice guidelines are important contributions to equitable healthcare, it is also crucial to document the actual clinical practices in hospitals that provide services to children with UHI to address points of improvement and future research needs.

### 1.4. Aims

Challenges associated with UHI have recently received increased attention, and children with UHI may benefit from early intervention. However, research is limited, and clinical practice may vary between countries and between clinics. Descriptions of the clinical practices related to the diagnostics and habilitation of children with UHI might uncover potential obstacles to equitable access to healthcare. Therefore, in this study, our aim is to describe the clinical practices for screening and diagnostic assessment of UHI while exploring the clinical considerations made for confirmed UHI in babies in Norway, Sweden and Finland.

## 2. Materials and Methods

### 2.1. Recruitment and Participants

All hospitals in Norway, Sweden and Finland that diagnose HI in children were invited to participate in the study. In total, 65 hospitals were invited. An online survey was sent to one contact person in each clinic, mostly a physician working within audiology. The person who received the survey was encouraged to discuss the questions with colleagues and, if applicable, to pass on the link to the survey to another colleague who might be better qualified to answer. The name of the hospital was asked to be reported in the responses. The identity and profession of the person(s) filling out the survey was unknown, but it was specified that the survey should be filled out by a person with clinical experience with newborn hearing screening and diagnostics. Participating hospitals were invited to a later online webinar about UHI as compensation for their contribution. Reminders were sent to non-respondents after two weeks and, in some cases, again later. Data from the three countries were then combined using only numeric codes referring to the hospitals that responded to the survey.

### 2.2. Survey Development

This study data was collected as part of the planning of a future longitudinal study on the impact of UHI on children’s development. A survey was constructed by the authors, including questions on screening, referral and diagnostic procedures, and whether the hospital had any specific procedures for newborns with a risk for HI. In addition, two short hypothetical case descriptions of suspected UHI were presented, followed by open-ended questions to allow for fuller descriptions of the current management and clinical decisions (Figure 1). The survey was translated from English into Norwegian, Swedish and Finnish. The translations were inspected by all the authors. Because many of the authors are fluent in at least two of the Nordic languages, it was possible to ensure that the Norwegian, Swedish and Finnish versions were comparable with each other.

### 2.3. Data Analysis

The collected data were partly suitable for a quantitative description and partly for qualitative analysis. The free-text responses provided the main part of the results and were subject to qualitative analysis. Quantitative data from the general questions regarding screening and diagnostic routines provided information that complemented the qualitative analysis. Thus, our study adheres to a qualitative dominant mixed methods design [53].

Table 1 presents information about the screening and diagnostic methods used. The responding hospitals vary regarding their catchment area population, with birth rates ranging from 200 to 28,900 per year. Therefore, to provide a more accurate estimate of the number of children affected by the various clinical decisions, each hospital’s responses were calculated as relative frequencies, that is, set in proportion to the annual birth rate of their catchment area. Statistical records of births in 2019 were used for this procedure [29,30,31]. In Finland and Norway, statistics for births per hospital were used, and in Sweden, births per region were used, as this was more accurate following the Swedish organisation of healthcare. 

The responses to the case description lent themselves to qualitative analysis. Content analysis was applied as described by Kyngäs [54]. The following procedure was used: One researcher from each country read thoroughly through all the written responses from their country several times to give an overall sense of the material. Next, the Finnish responses were translated into English, and one researcher (knowing Norwegian, Swedish and English) read through all the material and did the preliminary analyses. Each meaning unit was identified and labelled as a code describing the content. Then the codes were grouped and sorted into categories and subcategories based on the frequency of their appearance. The categories formed were then sorted into concepts and, finally, into main concepts. The coding was performed using a Microsoft Excel spreadsheet. 

The preliminary analysis was presented to the co-authors. Following discussion, adjustments to the main concepts were made until consensus was reached in the group. The data were analysed to identify patterns related to the current management of cases with UHI in the participating hospitals and hearing units.

## 3. Results

### 3.1. Respondents

Of the 65 invited hospitals, 61 responded to the survey. Some of these hospitals only did hearing screening and referred to other hospitals for diagnostic assessments and could therefore not provide answers regarding diagnostics and habilitation. Thus, 61 responses were received for hearing screening and 58 regarding diagnostics and habilitation. The Swedish results are presented per healthcare region. Each region mostly has one diagnostic hospital, but some have more than one. In Finland and Norway, responses were based on each hospital’s own clinical practice, which often covered a larger catchment area than the hospital’s own birth units, as children were referred to them from smaller hospitals in which HI in children was not diagnosed. Based on national and regional birth rate statistics from 2019 [29,30,31], the responding hospitals were estimated to provide HI diagnostics for 95% of the children born in Norway, Sweden and Finland. 

### 3.2. Screening and Diagnostic Methods

For the three countries, otoacoustic emissions (OAE) were used as the primary hearing screening method. In Norway and Sweden, both ears were always screened, whereas in Finland, only 19.4% of the newborns were screened in both ears. In Norway and Sweden, automated auditory brain stem responses (AABR) were mostly the primary method of choice as the first diagnostic test, and the threshold for normal hearing was mostly 35 dB in Norway and 30 dB in Sweden. In Finland, most hospitals used diagnostic auditory brain stem responses (diagnostic ABR) or auditory steady-state responses (ASSR) as the primary diagnostic method, and the most used thresholds for normal hearing were 30 or 20 dB (see Table 1 for details). Most hospitals in the three countries described extra procedures for babies at risk for HI, with AABR being the test most frequently mentioned and on-going monitoring for those who pass the screening.

### 3.3. Clinical Decisions for Habilitation

In the analysis of the responses to the two case vignettes, four main concepts were extracted: (1) hearing diagnostics and follow-up procedures of hearing, (2) conditional access to habilitation, (3) referral for other diagnostic investigations and (4) other habilitation actions and teamwork approaches within the hearing care systems. 

#### 3.3.1. Hearing Diagnostics and Follow-Up Procedures for Hearing

Excerpts from the responses:*‘Proceed with diagnostic ABR in natural sleep once possible’ (Case #1 and 2).**‘We shall make an appointment for a follow-up visit at the age of around five years when, as the primary form of examination, there will be pure tone and speech audiometry’ (Case #1).**‘Speech development would be monitored as a part of follow-up visits, speech therapy would be organised if needed, but, in this kind of hearing impairment, there is only seldom such a delay in speech development that would require speech therapy’ (Cases #1 and 2).*

Many hospitals aimed for an early diagnostic procedure when a child had a suspected UHI, mostly using ABR or ASSR. However, many hospitals, especially in Finland but also in Norway and Sweden, were quite reluctant to take early action. A few hospitals conducted a late follow-up visit at the age of five years for Case #1. After the confirmation of a HI, many hospitals reported regular hearing controls at least during the first years but less often or never in older children with UHI. One Finnish hospital reported yearly hearing controls and a follow-up protocol for the speech development. There was substantial variation in the follow up of the child’s healthy ear. Some reported that they did not check it at all, others reported on-going monitoring, and other responding hospitals did not mention a healthy ear.

#### 3.3.2. Conditional Access to Habilitation

Excerpts from the responses:*‘Hearing aids when the child can sit’ (Case #1);**‘No hearing aids before preschool’ (Case #1);**‘Regarding UHI in a newborn, we do NOT start any habilitation’ (Case #1 and 2);**‘For moderate and severe HI, hearing aids are important to stimulate the auditory nerve to enable cochlear implantation in the future’ (Case #2).*

Several responses emerged for treatment with hearing aids in children with UHI. Most responding hospitals would fit a hearing aid for Case #1 but only under specific preconditions. Besides the type and grade of HI, some hospitals would only fit a hearing aid if children had reached an age where they could sit by themselves, or if the parents were willing and motivated, or if the child responded well enough in specific audiometric tests. Some hospitals would fit a hearing aid, but not until the child was considered old enough, which had a spread between five months and preschool age (around 6 years). One hospital would not treat a child with contralateral routing of sound (CROS) hearing aids until the child reported a need to hear better in the classroom. In some cases, respondents stated that they would not provide individual parent guidance unless parents would ask for it themselves or when the children become older.

#### 3.3.3. Referral for Other Diagnostic Investigations

Excerpts from the responses: *‘It is important to check the vision when hearing is impaired’ (Case #2).*

In Norway and Sweden, the most common reason for referral to other clinics was to investigate the child’s vision. Other reasons were general health (paediatrics), malformation of ear, aetiological investigations such as congenital cytomegalovirus (cCMV) infection and genetics, and in Finland, a referral to speech and language therapist to investigate a possible language delay was common. Some responding hospitals did not include any information about referrals to other clinics or assessments besides hearing.

#### 3.3.4. Other Habilitation Actions and Teamwork Approaches

Excerpts from the responses: *‘If we provide hearing aids, they will be referred to audio-pedagogical unit for follow-up by a special education teacher or teacher of the deaf’ (Case #1).**‘We do not provide habilitation; we do not refer the child to speech and language therapist’ (Case #1).**‘Auditory Verbal Therapy (AVT) can be provided by a speech and language therapist or other professional with AVT training’ (Case #1).*

Some respondents, primarily Swedish hospitals, reported that children with UHI received similar intervention actions as children with bilateral HI. Others reported that families receive basic information or language testing at rare occasions and/or are invited to participate in parent courses. Some stated that no habilitation was provided for children with UHI. In the three countries, multidisciplinary approaches were described, organised in different ways. Swedish respondents described hospital-based teams comprising different professions, and in Finland speech and language therapists were often part of the hospital team. Some Norwegian hospitals typically referred to external service providers organised in other hospital units or as part of the education system, whereas others did not. Examples of professions mentioned were medical doctors trained in ear, nose and throat or audiology, speech and language therapists, teachers of the deaf, audiometricians, social workers and psychologists.

#### 3.3.5. Differences in Actions and Treatment between Cases #1 and 2

Many respondents described similar habilitation approaches for Cases #1 and 2, but there were also some specific differences. For instance, some hospitals (in all the three countries) would only investigate cCMV infection in Case #2, in which single-sided deafness (SSD) was suspected but not in Case #1 that suggested moderate UHI. Only a few hospitals investigated cCMV in both cases. One hospital stated that they would refer a child with SSD to a cochlear implant (CI) investigation only if a child had been diagnosed with a congenital cCMV infection. Most of the responding hospitals did not mention cCMV in their responses. There were also some differences regarding the provision of hearing aids. Many hospitals suggested hearing aids for moderate UHI, although some had specific conditions, as previously mentioned. In the case presenting SSD, most hospitals would not fit a hearing aid but would rather consider referral to CI investigation.

#### 3.3.6. Variation across Countries and Regions

Overall, the Finnish hospitals differed somewhat from the Swedish and Norwegian hospitals regarding screening procedure, as they mainly screened only one ear. Compared to Sweden and Norway, there was a tendency of a delay in timing of diagnostics and habilitation. However, as the variation in content and timing of interventions was considerable in all three countries, the within-country variation was more pronounced than between countries. The timing and content of the interventions provided by the hospitals did not seem to depend on the hospital size.

## 4. Discussion

Our study revealed considerable variations in organisation and clinical practices for Nordic children with UHI, both between countries and within the same country. Thus, access and content of early intervention may vary across hospitals from no intervention at all—in some cases, not even a diagnosis—to more comprehensive, multi-disciplinary intervention in other cases. As UHI might impact developmental areas such as auditory skills, language and quality of life, as some studies suggest [19,20,23], the heterogeneity in service provision means that some families might not receive the appropriate support. 

Our data confirm figures from a previous report [48] that mention Finland as one of the few European countries that screen only one ear in newborns. This procedure might be due to few resources in the hearing health care, or the assumption that early detection of UHI is not urgent, as firm evidence is lacking. Therefore, many Finnish children with UHI might not be diagnosed following screening and will thus miss the possibility of early intervention and monitoring of early development. However, the expected association between screening routines and prevalence numbers is not confirmed, as recent prevalence statistics for UHI are not available for Finland. 

Our results also revealed variations in screening and diagnostics regarding access to early intervention and the types of interventions that were provided. The heterogeneity of services is not surprising, given the limited evidence base available, and similar results have been reported by others [27,49,50]. Even if heterogeneity in early intervention can be expected because each child and family has different needs, the variation between hospitals suggests that families receive quite a different type of intervention depending on where they live. Furthermore, the early intervention systems are organised in different ways, and the hospitals that refer families to external institutions for parts of the intervention (e.g., speech and language therapy) probably could not provide accurate descriptions of everything available in their area.

As the responding hospitals varied greatly in size, with annual births ranging from 200 to 28,900 in their catchment area, the interventions might depend considerably on resources and competence available for each clinic. The present study did not demonstrate any association between hospital size and habilitation routines, albeit the study was not designed to detect such associations and the use of a mainly qualitative approach limits the possibility to explore this question. Future studies that address the resources and competence available for hospitals might be useful for future policy making and healthcare planning.

Some of the responding hospitals would provide amplification early to a child with UHI, whereas others would do this later, or not at all. This variety matches other reports of clinicians’ and parents’ decisions regarding amplification in children with UHI [50,55]. Various criteria for amplification were mentioned, such as type and degree of HI, parents’ or children’s motivation or the child’s age or developmental level. It is not known whether these criteria are based on evidence, clinical practice or general opinions; however, the variation between hospitals suggests a lack of equitable access to services in all three countries. As for degree, hearing aids were mostly recommended regarding moderate UHI and not SSD. This corroborates existing recommendations on UHI management [6,36], suggesting that with moderate UHI, the affected ear can be aided with a hearing aid, whereas profound degrees of UHI would not be aided and other interventions, such as CI or remote microphone systems, should be considered [6]. 

The consideration of the parents’ and the child’s opinions is central for patient involvement and shared decision making. However, the informed opinion of a parent presupposes access to adequate, unbiased information [56], and there is a risk that parents or children might be unaware of the potential benefits of amplification. Therefore, it is essential that health care professionals are aware of the options and that they acknowledge their responsibility regarding parent information. Parents have confidence in professionals’ recommendations, but they could also be confusing if professionals provide contradictory advice, thus making decision making difficult [55]. Children with UHI function seemingly well in many situations, and it might be challenging for a parent to detect any need for amplification. As previously mentioned, there is some evidence of improved binaural hearing if amplification is provided at an early age, and delayed amplification might hamper this development [46].

We presented two cases for the hospitals: one in which a moderate UHI was suspected (Case #1) and another with a suspected SSD (Case #2). Many hospitals described similar procedures in the two cases. Some, however, would test for cCMV infection if SSD was suspected but not for UHI. Given that cCMV infection is associated with any degree of HI [57], it is recommended that all children who fail the screening are screened for cCMV [58]. Thus, this should apply to children with any degree of UHI and those with SSD. However, the awareness of cCMV infection is quite recent, and the routines for screening and treatment probably remain under development in many hospitals [59].

### Strengths and Limitations

Our survey offers insight into the clinical practice in Norway, Sweden and Finland regarding screening, diagnostics and habilitation of newborns with UHI using quantitative descriptions and qualitative analysis.

By collecting data from clinicians in the form of free-text responses to two hypothetical case descriptions, we could provide information about what is performed and the underpinnings of the clinical judgement for children with UHI. Whereas this is valuable information, the chosen method has some limitations requiring consideration. We have not investigated patient registers for providing exact numbers and procedures regarding children with UHI, and the free text format of responses also implies that the lack of a specific intervention among the responses does not necessarily mean that it is not provided; some parts of the service provision might have slipped from memory, or they may be provided by other institutions and therefore not recognised by the hospital staff. As the identity of the respondents are unknown, there is also a risk that the questions in the survey were misunderstood if the respondent did not have audiological expertise. However, based on our knowledge of how audiological services are organized in the Nordic countries it is assumed that respondents had audiological training and were familiar with the terminology. Although care was taken to ensure that the translation of the survey questions was comparable in content in Norwegian, Swedish and Finnish, there may still be differences in meaning in the three languages that prompted slightly different responses. 

## 5. Conclusions

As Norway, Sweden and Finland are high-income countries where healthcare is considered accessible for all inhabitants, the variation in timing and content of services might seem unexpected. However, the responding hospitals vary greatly regarding the clinic and population sizes they serve, and we cannot rule out that limited resources might be an issue in some hospitals. More importantly, however, given the scarcity of research, the variability might simply reflect different clinical opinions rather than access to resources. Given the state of knowledge, it is reasonable to assume that early detection and intervention are important for children with UHI, as the condition is associated with adverse developmental consequences and may also progress to bilateral HI over time. Still, best practices regarding the scope and content of early intervention for this group remain unestablished. Guidelines and controls of the hospitals’ compliance to the guidelines are warranted. 

Thus, there are two potential pitfalls we need to avoid: late diagnosis and limited access to early intervention, which might have severe consequences for children who need early intervention, such as children with additional health conditions or children from families with low socioeconomic level. Concurrently, some interventions are costly regarding families’ and service providers’ resources and should not be provided unless the benefit is well documented. The severity of these scenarios suggests that research addressing the development of early-identified children with UHI and the appropriate timing and content of early intervention for this group is imperative. 

## Figures and Tables

**Figure 1 jcm-10-05152-f001:**
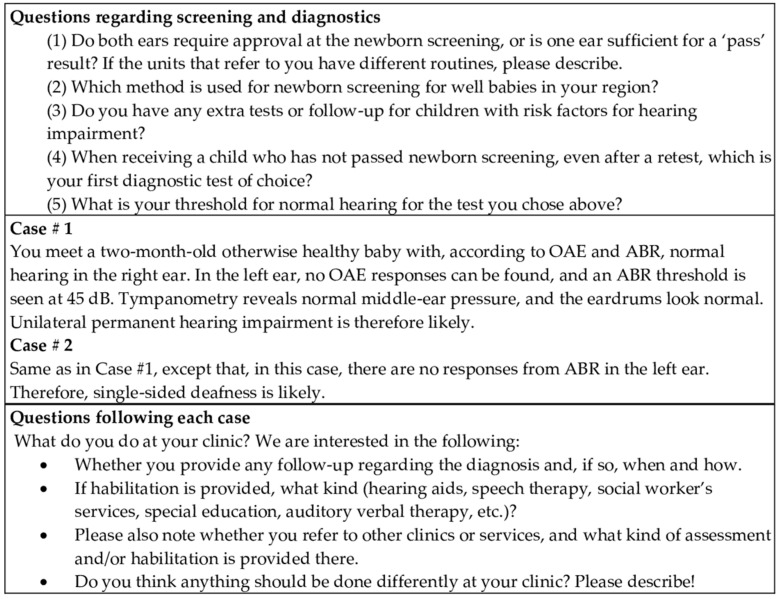
Questions and case vignettes presented in the survey.

**Table 1 jcm-10-05152-t001:** Results from the questionnaire in percentages by childbirth coverage.

Survey Item	Norway	Sweden	Finland	Total
**Number of Hospitals**	15	29	17	
**SCREENING**				
**Requires Pass in Both Ears**	100	100	19.4	83.6
**Screening Method for Well Babies**				
OAE	100	100	100	100
**Extra Tests for Babies at Risk**				
Yes	87.2	97.0	79.3	92.3
No	12.8	3.0	3.8	4.9
Missing	0	0	16.9	2.8
**DIAGNOSTICS**				
**First Diagnostic Method of Choice**				
Automated ABR (AABR)	90.5	98.8	0	75.2
Diagnostic ABR	3.6	1.2	54.1	13.4
ASSR	0	0	43.5	9.5
Other	5.9	0	2.4	1.9
**Threshold for Normal Hearing**				
20 dB	0	0	44.3	9.7
25 dB	16.5	0	6.0	5.1
30 dB	6.7	66.9	47.4	48.7
35 dB	76.7	30.1	0	34.3
Missing	0	3.0	2.4	2.2

Note. OAE, otoacoustic emissions. ABR, auditory brainstem response. ASSR, auditory steady-state response. dB, decibel.

## Data Availability

Due to the nature of this research, participants of this study did not agree that their data would be shared publicly; thus, supporting data is not available.

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
