# Peer review of "Newborn Hearing Screening and Intervention in Children with Unilateral Hearing Impairment: Clinical Practices in Three Nordic Countries"

_jcm, 2021, doi:10.3390/jcm10215152_

Round 1

Reviewer 1 Report

Review - Journal of Clinical Medicine

Newborn Hearing Screening and Intervention in Children with Unilateral Hearing Impairment: Clinical Practices in Three Nordic Countries

Overall Impressions:

  1. The topic of this paper is likely to be of interest to readers of this special edition of JCM
  2. On the whole, this manuscript is well-written, clearly organized, and informative
  3. The topic provides useful information to the field and a lens into how various countries conceptualize, address, and support children with unilateral hearing loss and their families
  4. The resources are well placed in the text and include relevant and important citations from this field of inquiry
  5. The use of the term “unilateral hearing impairment” may prove somewhat problematic in terms of acceptability of this work for an international audience. Many journal specific to this population advise against the inclusion of “hearing impairment” and instead promote “hearing loss” or “hearing levels.” To this reviewer’s knowledge, this is not the case for the present journal; thus, this terminology may be acceptable in the present paper. However, the authors are encouraged to consider whether they might be willing to adjust the terminology to make their work more palatable to an international audience.
  6. The Methods section is not as robust as the remainder of the paper. Attention should be given to strengthening this section to ensure that readers understand the steps the authors have taken to arrive at their conclusions.

More specific feedback:

  1. Introduction
    1. The Introduction does a nice job of succinctly articulated the research that positions this paper
    2. It would be helpful to have more clarity regarding the implementation of universal newborn hearing screening across the three nations. Was the rollout of these efforts uniform in each country (e.g., On a particular date, the nation started doing it)? Or is the 2001-2008 range because implementation started in urban areas? Was the “roll out” of the screening slower to reach rural areas or areas with a greater catchment area? Adding a sentence or two here to clarify would be helpful to the reader.
    3. The authors made good points about the variability of implementation of Early Intervention across other contexts and highlight the lack of research about EI efficacy (beyond audiologic outcomes).
  2. Materials and Methods
    1. Congratulations to the authors on their planned longitudinal study on the impacts of unilateral hearing differences on children’s development. This work is well situated to make a substantive impact on the field.
    2. Where the surveys translated AND back translated? If so, say that. If not, how did the authors ensure that the translations were high quality?
    3. Did the researchers ask for people in a particular role to answer the question, preferentially? It is understood that the survey could be completed by any person deemed most appropriate by each hospital. However, it remains relevant to identify the types/roles of the persons to whom these were addressed.
      1. Individuals in what type of role(s) comprised the majority of recipients of the surveys? Audiologists? Otolaryngologists? Cochlear implant team members?
      2. The questions assume audiologic background (e.g. understanding what “no OAE responses” mean) – if someone without that foundational knowledge responded, the results might be skewed
    4. The authors are encouraged to provide more information about their qualitative methods & content analysis approach.
      1. Was one researcher in each country charged with the coding?
      2. What mechanisms of inter-rater reliability were considered or utilized?
  • Did the authors from each nation check each other’s coding system?
  1. How were differences in coding approaches addressed, if they existed?
  2. Was a coding system/software used?
  1. The authors describe the methods as part qualitative, part quantitative. Might a mixed methods approach be used to describe this work? Are the two portions of the study conceived of as separate entities? How did findings from one influence the other? More clarification regarding the Methods are necessary.  
  1. Results
    1. 58/65 hospital responses regarding diagnosis and habilitation is quite good. Congratulations to the researchers on their successful inclusion of so many participating institutions.
    2. The descriptive study does not need to delve into the intricacies of why just 19.4% of newborns in Finland are screened in both ears. However, interested readers may be perplexed by this. Adding a statement, even parenthetically, about why this approach is taken in that context could be helpful.
    3. Under 3.2, pg 6, line 232 – please describe what is meant by “regular hearing controls for those who pass the screening.” In a study like this, “regular hearing controls” would usually refer to children with typical hearing, in contrast to/as a control group for children who have reduced hearing. Perhaps this refers to “on-going monitoring”? Please describe.
    4. Where there any differences between countries in their “endorsement” of the four main concepts? (E.g., in Norway 50% of responses were related to #1, whereas in Sweden, just 20% of responses were focused on #1)? It may not be necessary to indicate this with percentages; however, letting the reader know, for example, “These were equally endorsed across the three countries” or “All three countries shared similar responses for #2, yet there responses for #4 varied widely” could be helpful. To what extent did these Nordic nations “hang together” and/or differ in their responses?
    5. 3.1 – “reported regular controlling of hearing” – this is likely related to ‘c’ above. The meaning of this is not clear to this reviewer.
    6. 3.2 – the findings in this section were surprising to the review. These are relevant and important to include to provide a context for why this study is important.
  2. Discussion
    1. The authors are encouraged to ‘circle back’ to a few of the comments made in the Introduction about the potential negative outcomes that can be associated with reduced unilateral hearing. If those who ‘skim’ the paper go to the Discussion, it might be worth adding a sentence or two reiterating the consequences of untreated/unsupported.
    2. The points made about parental involvement and investment in decision-making are solid. It is true that caregivers need to be made aware of their options; it also is true that medical health providers need to be made aware of options for this population of children as well.
  3. Conclusion
    1. The point about unilateral progressing to bilateral hearing differences is an important one.
    2. The authors are encouraged to expand upon “…limited access to early intervention might have severe consequences for children who need early intervention [such as ….] – although “best practices” are not fully developed, there are known risk factors that make early intervention more likely to be beneficial.

Thank you for the opportunity to review this paper. This reviewer wishes the authors the best of luck in their future collaborative research endeavors.

Reviewer 2 Report

This paper focus on unilateral congenital hearing impairment (UHI) which is quite common with a prevalence of about 0.5-1:1000 newborn. Still the impact for children with UHI is quite unknown and the best diagnostic procedures and habilitation are discussed. As a basis for further studies, the authors evaluated the screening and diagnostic procedure in Norway, Sweden and Finland.

The article presents a good overview about the topic UHI with many recent references.

 In my opinion, the introduction is much too long and detailed and should focus more on the main topics of this paper. For example in chapter 1.1. and 1.3. it might be enough to know, that the impact for children of an UHI is discussed controversely and definitions and guidelines are rare; line 72-83, 127-137, 142-148…. are not the focus of this paper. Some parts are redundant (i.e. line 127 and 155).

 Even though the responses to the case reports had a manly qualitative approach, it would be also interesting to have a table with some information for the different countries about the diagnostic and habilitation process. I would suggest to split table 1 in one table for screening (first three lines) and second for diagnostic process/habilitation and complete it with i.e. a percentage how many clinics provide follow-up at which age, if and what habilitation is provided in  case 1 and case 2, mention of cCMV…

As it is discussed that the results might depend on the size of the hospitals, it could be a good idea to further split up table 1 (and the table about diagnostic procedures/habilitation) by size of the reached population (i.e. small, (middle), large).

It also would be interesting if the prevalence of UHI in Finland is lower (as you might think with the screening of only one ear) and how many hospitals fulfil the Norwegian guidelines in this country.

One conclusion might be that guidelines, documentation and regular evaluation are needed in these countries to standardize the processes.

Line 53/54: ….UHI in adults as a pure tone average (PTA) of hearing thresholds at 0.5,1,2 and 4 kHz.

139-142 should be removed to the discussion as it is repeated in line 323-326

186: Table 1 (not 2)

211-213: sentence

Table 1: number (N) of hospitals for each country,

“passed ears required in the screening”: percentage for one OR both is enough as the sum is 100%

275: Abbreviation CROS is not needed as it is not used again

294: abbreviation AVT?

Reviewer 3 Report

The subject is actual. The early identification of congenital unilateral hearing impairment (UHI) raised significantly with the implementation of newborn hearing screening. There is a need for guidance for intervention and rehabilitation strategies in these babies.

The Introduction the definition of UHI and data on its’ prevalence are presented. National healthcare systems are described thoroughly. The main issue is to define outcomes measures taking into account methodological limitations revealed in previous studies. The study aims to observe the current practices in the management of babies with UHI by the mean of hospitals survey.

The style and structure of the article meet the requirements. The results are presented clearly. The data analysis is based on correct qualitative and quantitative methods.

Author Response

The authors thank Reviewer 3 for the comments. We have not done any revisions based on this review as none were requested.

Round 2

Reviewer 2 Report

Thank you for the revision of the paper - I think it is now much better, more structured and gives more information about the differences in the three countries, as far as this is possible with the qualitative approach.